# African Swine Fever—How to Unravel Fake News in Veterinary Medicine

**DOI:** 10.3390/ani12050656

**Published:** 2022-03-05

**Authors:** Adriana Trotta, Mariarosaria Marinaro, Alessandra Cavalli, Marco Cordisco, Angela Piperis, Canio Buonavoglia, Marialaura Corrente

**Affiliations:** 1Department of Veterinary Medicine, University of Bari “Aldo Moro”, Str. Prov. per Casamassima Km 3, 70010 Valenzano, Italy; alessandra.cavalli@uniba.it (A.C.); marco.cordisco@uniba.it (M.C.); a.piperis2@studenti.uniba.it (A.P.); canio.buonavoglia@uniba.it (C.B.); marialaura.corrente@uniba.it (M.C.); 2Department of Infectious Diseases, Istituto Superiore di Sanità, Viale Regina Elena 299, 00161 Rome, Italy; mariarosaria.marinaro@iss.it

**Keywords:** mass media, viral infections, infectious disease, wildlife, conspiracy theories, quackery

## Abstract

**Simple Summary:**

In recent years, fake scientific news has spread much faster through the Internet and social media within the so-called “infodemic”. African swine fever (ASF) is a perfect case study to prove how fake news can undermine the public health response, even in the veterinary field. ASF is a contagious infective disease exclusively affecting domestic and wild pigs such as wild boars. ASF can cause social damages and economic losses both directly (due to the high mortality rate) and indirectly (due to international sanctions). Although ASF is not a threat to human health, since 2018, newspapers have often reported false or misleading news, ranging from misinterpreted findings/data to fake or alarmistic news. In some cases, fake news was spread, such as the use of snipers at the border of nations to kill wild boars or the possible risks to human health. In order to provide real and fact-based news on epidemics, some organizations have created easy-to-read infographic and iconographic materials, available on their websites, to help the readers identifying the fake news.

**Abstract:**

In recent years, fake scientific news has spread much faster through the Internet and social media within the so-called “infodemic”. African Swine Fever (ASF) is a perfect case study to prove how fake news can undermine the public health response, even in the veterinary field. ASF is a highly contagious infective disease affecting exclusively domestic and wild pigs such as wild boars. ASF can cause social damage and economic losses both directly (due to the high mortality rate) and indirectly (due to international sanctions). Although ASF is not a threat to human health, since 2018 newspapers have often reported false or misleading news, ranging from misinterpreted findings/data to fake or alarmistic news. In some cases, fake news was spread, such as the use of snipers at the border of nations to kill wild boars, or those reports concerning possible risks to human health. In order to provide real and fact-based news on epidemics, some organizations have created easy-to-read infographic and iconographic materials, available on their websites, to help the readers identifying the fake news. Indeed, it is crucial that governments and scientific organizations work against fear and anxiety, using simple and clear communication.

## 1. Introduction

The Internet and social media have helped to spread news and information by breaking down geographical and temporal barriers, allowing people to be constantly updated on any topic [1]. Mass media, especially social networks [2,3], deliver a huge amount of information daily, even from unreliable sources [1,3,4,5,6]. In fact, according to a recent statistic, about 4.54 billion people (59% of the global population), around the world, are active internet users, and obtain information using websites and applications of publishers [6]. When the source is mediated by an algorithm, the digital media platforms act as intermediaries and readers randomly access the news [7], without knowing the source [5]. This happens especially for scientific news when a specific background is required to completely understand the topic, whether in the medical or veterinary field [8,9,10]. In recent years, as it has been observed during the COVID-19 pandemic, the number and popularity of conspiracy theories, scientific quackeries, pseudoscience and alternative ‘facts’ have increased steadily, especially online [3,11,12,13,14]. In fact, the advent of the Internet has rendered scientific papers easily accessible to anyone and has led, for example, to the dangerous practices of self-diagnosis and self-medication [15].

Science topics are often difficult to explain to a large audience and must therefore be simplified for mass communication [16]. Moreover, modern society seems to have lost faith in the scientific community [17,18]. A distinction is necessary between incorrect news that is accidentally disseminated (“misinformation”) or badly received by the reader, and news that is intentionally distorted (“disinformation”) [19]. Both types can be considered fake news, but the scope, and therefore the consequences, are different [20]. An appreciable difference between casual and intentional disinformation relies on the attempt to proselytize someone, which is frequent in the second case. Readers who embrace the disinformation believe that health policies are not written by scientists but by “conspirators who act in secret for their own personal gain, and not for the interests of all and the public health” [17]. This misbelief pushes proselytes to merge, guided by a group leader who often acts for her or her own personal benefit [21]. Moreover, those who believe in fake news usually spread it intentionally, allowing this news to have a boomerang effect [22]. An interesting case, and one which is discussed here, is the African swine fever (ASF) epidemic, which is an example on how fake news in the veterinary field has indeed had an impact on public health. In particular, this paper aims to describe the specific pseudoscientific news spread regarding the ASF epidemic and also presents easy-to-access websites, blogs and papers useful to unveil nonscientific sources and how to counteract them.

## 2. The Case of ASF Outbreak

The causative agent of ASF is the African swine fever virus (ASFV), a large double-stranded DNA virus which can infect several members of the *Suidae* family, and is the sole species within the *Asfivirus* genus, family *Asfarviridae* [23].

The ASFV is very stable and shows a very low mutation rate; in addition, the absence of related viruses makes recombination events very unlikely; thus, the risk for ASFV to jump the species barrier is not significant [24]. To date, 24 ASFV genotypes have been described, based on the sequence of the gene encoding the capsid protein P72 [25], and the genotype II is responsible for the recent pig pandemic, which is still ongoing [26]. ASF is a contagious disease that affects domestic and wild pigs, but it is not a zoonosis, and the ASFV cannot infect other mammalian families. Susceptible subjects can become infected via nasal, oral, subcutaneous, or ocular routes, and can shed the virus into the environment even after death, since the ASFV is highly persistent in carcasses and can be found in raw pork products [27]. In addition, the virus can persist in feces and urine up to 8 and 15 days, respectively, and for 180 days in processed products [28]. In Europe, the ecology of the disease is characterized by both direct and indirect transmission [29] and, in some geographic areas, where pig breeding is wild or semi-wild, the “boar cycle” allows the virus to spread from domestic pigs to wild species [30]. The wild boars can have a role in the transboundary spread of the disease moving across borders of nations [30,31,32,33]. Another prominent cause of incursion of ASFV in ASF-free areas is the importation of infected animals or infected pork products, although global legislations on animal health, ban the animal trade from high-risk areas.

ASF is an ongoing pandemic and an economic emergency as well, since: (i) it can easily spread to pigs in disease-naïve areas and some viral strains have a mortality rate of 100% [28]; (ii) the economic losses can be huge especially in free-disease regions where the disease has a devastating impact on the swine livestock sector [28] due to culling of animals and movement restrictions of animals (and their products) in order to limit the spread of the infection/disease to neighboring areas or to other countries [34]. Of note, although several evidence indicate that pigs recovered from infection are able to produce antibodies, there is still no commercial vaccine available for ASF [35].

China and Vietnam are countries where the ASF epidemic has hit hard. The ASF has a particularly strong economic impact on smallholders and commercial farmers, since pig meat is a major source of protein in human diets, and the livelihoods of numerous households, which depend on pigs as a source of protein and income, are at risk in several Asian countries [36,37].

Historically, the disease was present in the African continent but after entering Portugal, in 1957 and again in 1960, it became endemic in the Iberian Peninsula, and several linked outbreaks were observed in some European countries in the following years [30,34]. After these outbreaks, ASF was considered completely eradicated in Europe [28,34] except for the outbreak occurred in Sardinia (Italy) where the disease appeared in 1978 and it is still under eradication, which is expected to be achieved in a few years [38,39]. In recent decades, the epidemiology of ASF has rapidly changed; indeed, the disease is now present in wild and/or domestic pigs in several regions of Asia, Europe and Africa [40,41,42,43,44]. In particular, the disease reached China and Vietnam in 2018–2019 and was notified in Germany in 2020 and in Italy in 2022 [44].

## 3. ASF and Fake News

Due to recent new outbreaks, fake news, quackeries and conspiracy theories have been spread via social networks and the Internet on ASF and ASFV.

### 3.1. ASF in China

ASF outbreaks and restriction measures have caused huge economic losses in China, due to the enormous consumption of pork meat per capita [35]. Thus, in August 2018, when ASF appeared, the social media and some nonscientific newspapers began to spread true information in a catastrophic way and, finally, they ended up releasing fake news.

The example reported following is the news published on the UK online newspaper Express, with the title “Chinese New Year celebrations ‘could spark DEADLY swine fever OUTBREAK’—shock warning” [45]. The scope was clearly to create alarmism also by using both capital and bold letters for the words “deadly” and “outbreak”.

### 3.2. ASF in South Korea

In 2018, ASF arrived at the gates of South Korea and an Italian journal, “Il Giornale.it”, published an article with the following title: “South Korea, snipers on the border to kill wild boars” [46]. The paper, which mentioned the wide deployment of drones and the use of armed force to spot and kill wild animals, referred both to the South Korean government’s fear to import disease from North Korea and to the difficulty of implementing the containment measures to control the ASF epidemic.

In a very short time, it was clear that such containment measures were insufficient to control the outbreak and, one year later (12 November 2019), another Italian newspaper “Il Messaggero.it” titled “The river turns red: it is the blood of 47 thousand slaughtered pigs”, with an eloquent and dramatic picture of a huge river whose water had turned red [47]. The news was also reported by numerous international newspapers through the web [48,49,50]. All these papers emphasized the contamination of a South Korean River with the blood of slaughtered pigs. However, this event was unreal and, in fact, searching on the web, it was possible to find the same picture posted in the past, when industrial spills caused the change of the color [51].

### 3.3. ASF across ASIA

In 2019, The Guardian published an article with the following title “No way to stop it: millions of pigs culled across Asia as swine fever spreads”, with a more catastrophic subheading stating “Experts say region is losing the battle to stop the biggest animal disease outbreak the planet has ever faced” [52]. The article had the clear intent to generate anxiety in the readers, especially when stating that a “battle” was “lost” and when comparing the ASF to the Ebola epidemic. Furthermore, the names of the “experts” who made such a comparison were omitted.

### 3.4. ASF in Europe

In 2018, when ASF reached Europe, the news was promptly reported by online sources: “African swine fever, fear in Europe: outbreak in Bulgaria, euthanasia of infected pigs” [53]. The journalist in this case wanted to underline the “sacrifice of animals”, using the word “euthanasia”, which in this context sounds inappropriate. In addition, the article incorrectly stated that the “current outbreak comes from Africa”, ignoring the epidemiology of ASF in Europe.

Far more serious and alarmistic was the news reported by an international newspaper whose headline stated “France deploys ARMY in ferocious battle against WILD BOAR amid swine flu outbreak” [54]. The news echoed the story of soldiers at the country borders ready to avoid the entry of infected wild boars (underlining how costly this measure was for France). The author released false news, in an alarmistic and misleading way, reassuring the readers that the use of the armed force would only support hunters (hired to place traps). Therefore, and as it often happens for newspapers that report fake news in their titles, when reading the entire article, the readers could find out that the initial claim was denied.

In addition, in this title, the word “flu” was used improperly, and this is another typical misleading news that it is repeated by the media about the ASF disease, with “swine flu” and “swine fever” considered interchangeable. This is totally incorrect, as they refer to different diseases, sustained by different infectious agents.

### 3.5. Fake News about Human Risks

When the ASF reached the Eastern European farms, the online media continued to spread fake news, fueling the confusion of the readers on the potential zoonotic role of the ASFV. Often, in the search for an effective title, this hypothesis is launched, and it is systematically refuted later in the article. For example, the Italian newspaper “Il Fatto Quotidiano” published the following article on 26 July 2019: “African swine fever is epidemic. The risks for humans and the absence of a vaccine” [55]. The title caused anxiety and uncertainty about the risks for humans, and warned even more, by mentioning the lack of a vaccine. The reader needed to read the full article to find out that humans are not susceptible to the disease and that there are no vaccines available to prevent disease in pigs.

The possibility that ASFV could spread from pigs to humans was also reported in the 2019 article “African swine fever: What is African swine fever—and could it infect YOU?” [56], in which the author stated that “One research paper authored in the American Society for Microbiology’s Journal of Virology discovered other viruses in the group which could infect humans…The researchers identified a strain of the virus which has been observed in human samples”. The strategy of citing a scientific paper to give strength and authority to their own thesis is a leitmotif in newspapers reporting fake news as they often use the tricks to incorrectly cite the source or make it impossible for the reader to find it.

In addition, in the Guardian “What is African swine fever and how does it spread?” [57], the authors state that “Humans cannot contract ASF. However, the head of the Russian epidemiology service, … has warned that pig physiology is close to human physiology, and that future mutations of the virus may therefore become dangerous to human beings too”, therefore warning on possible deadly mutations, which is another typical claim of those spreading fake news on infectious diseases.

Finally, in the Italian and in the Spanish languages, ASF disease is commonly translated as “peste suina africana” and “peste porcina africana”, respectively, i.e., “African swine plague”. Since in both languages the word “plague” is improperly used, this definition is likely to mislead the public, who believes that ASF is similar to the well-known zoonotic diseases caused by Yersinia pestis, a bacterium responsible for the disease called “bubonic plague”, i.e., the Black Death.

### 3.6. Conspiracy Theories

During an outbreak, conspiracy theories offer people simple explanations for complex problems; however, they place the blame on others who are often demonized [58]. Those who put forward and spread conspiracy theories often see themselves as victims and truth-sayers, and are firmly convinced that they are right [59]. Conspiracy theories about threats to human life, such as diffusion/dissemination of fatal viruses or outbreaks, are effective in sowing mistrust, confusion, and fear, and throughout history they have been used to control readers who do not have sufficient scientific knowledge [12]. Furthermore, conspiracy theories regarding the 1889–1892 Russian Flu, the 1918–1920 Spanish Flu and the actual 2020–2021 COVID-19 pandemics, are eloquent examples [12].

An interesting conspiracy theory was spread online, as denounced by the Center for European Policy Analysis (CEPA), through the thesis “Kremlin media poisons Baltic debate about African virus”, available in the website “Viral Disinfo” [60]. The paper denounced that the Sputniknews.it wrote about the ASFV outbreak in Lithuania and other two Baltic states (“the disease was caused by the Pentagon working out its strategy and tactics of biological war”) and the article referred even to a possible origin of the virus in the laboratory because of the “resilience to the cold Northern climate” which “proves that it was developed in a biological laboratory and that the U.S. military bio-lab Fort Detrick is involved because offensive infection agents are developed there”. Again, the Rubaltic.ru ping-ponged the subject and claimed that “the U.S. is testing biological weapons in the Baltics and is preparing for a Third World War”.

Similar conspiracy theories were already spread in the 1970s, e.g., “CIA brought ASFV to Cuba to undermine the Cuban economy and the Castro regime”, and again, in 1980, when the quackeries about the HIV/AIDS virus suggested that the HIV was created by the CIA under the executive order of the U.S. President Richard Nixon, to wipe out homosexuals and African Americans. Rumors and fake news could therefore be considered as part of the disinformation during the Cold War [61].

The potential link between HIV and ASFV has even been discussed by Feorino and co-authors in 1986, with the letter entitled “Aids and African swine fever virus”, which rejected the false rumors regarding this possibility [62].

## 4. Fighting Fake News

Fake news released from mass media can be considered as a “virus” that spreads among all those who are subjected to the disinformation [19], and recently, during the COVID-19 crisis, the World Health Organization (WHO) has stated that the world is not “just fighting an epidemic; we’re fighting an infodemic”, caused by the propagation of conspiracy theories, and misinformation about the pandemic [12]. In fact, often the solutions to this problem are similar to “antivirus programs for computers”, which have the aim of identifying the online primary source of the false news to block it, before other users become “infected” [61]. For this purpose, the ECDC has developed the R-based tool epitweetr as part of the “Epidemic Intelligence project”, which “allows users to automatically monitor trends of tweets by time, place and topic, with the aim of detecting public health threats early through signals, such as an unusual increase in the number of tweets” [63]. An additional question to be addressed is: how could public health authorities act constructively to fight the emergence of both conspiracy theories and false news? During the “online infodemic”, some responses came from official websites of governments and scientific organizations (WHO, ECDC, OIE, FAO, etc.), whose primary role is to disseminate correct scientific information and to provide useful tools to identify false news. For this purpose, on the European Commission and UNESCO websites, a set of ten educational infographics, have been published, with the aim of helping the public in identifying, debunking and counteracting conspiracy theories [64]. Moreover, researchers have created numerous national and international organizations with the purpose to diffuse correct scientific information and to fight against scientific quackeries. Among these, “Patto per la scienza” [65], the Pan American Health Organization (PAHO), which created the website www.paho.org, with special contents to counteract the Infodemic [66], and the Comparative Analysis of Conspiracy Theories (COMPACT) organization [67], which has the aim of thwarting scientific fake news and conspiracy theories through a network of academic researchers from 35 countries across Europe. The COMPACT has also released a short guide to fight conspiracy theories, along with recommendations of how to deal with them, which can be downloaded for free [68]. In particular, for fake news related to ASF, the “Office International des epizooties” (OIE) has published an extensive list of “myth-busters”, endeavoring to provide counter-information on important issues, through the creation of some easy-to-approach images [69]. In addition, a webinar dedicated to ASF, and the need for correct communication campaigns, was organized. The webinar is still available on YouTube [70]. Furthermore, the activity of the working group of the GF TADs network (Global framework official site) [71] has been intensified. This site has the advantage of offering informative materials, in several languages (English, French, Russian, Mandarin, Spanish), adapted to different local situations to provide useful information to travelers, and explains also all the biosecurity measures to be adopted by farmers, during animal transportation and at the slaughterhouse, in case ASF is suspected. The EFSA organization, in addition to publishing a detailed and updated report [72], has released an animated video available on YouTube [73] to clarify the epidemiology of ASF.

## 5. Conclusions and Recommendations

The Internet and social media provide enormous advantages but also serious disadvantages to scientists, journalists and readers. In the last few years, social media such as Twitter or Facebook have become immensely popular even in the science communication; therefore, scientists have started to use them, opening their research to a broader public [3]; however, social networks are also full of pseudoscientists who can have numerous followers. Therefore, to unravel fake news, the following are is necessary: (i) scientists should adopt the scientific method to verify the messages spread by pseudoscientists [3]; (ii) journalists should adhere to standard ethical principles and only disclose news when received from reliable sources; (iii) the public should be trained to interpret the news they read. For this purpose, it is important that governments and scientific organizations work against fear and anxiety [74], using simple, clear and comprehensive language, since it is known that false claims can delay treatment-seeking and promote reckless behavior [75]. The “viral” spread of misinformation by harmful pseudoscientific newspapers should be intercepted in a timely manner, and stopped without hesitation, especially on social media [3], where false claims originate and propagate rapidly. The lessons from past and ongoing pandemics could help manage and counteract future infodemics.

## Data Availability

The data presented in this study are available in the paper.

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
