# Peer review of "African Swine Fever—How to Unravel Fake News in Veterinary Medicine"

_animals, 2022, doi:10.3390/ani12050656_

Round 1
Reviewer 1 Report
The Authors with the Commentary entitled "African swine fever and its natural hosts - A case study to un-ravel fake news in veterinary medicine", provide real and fact-based news on ASF, and critically elucidated the disadvantages created by internet and social media to scientists, journalists and readers, especially all fake news about human risks. In fact, the African Swine Fever (ASF), as reported by the authors, is a perfect case study to prove how fake news can undermine the public health response even in the veterinary field.
The concept of the improperly used language and the misleading word “plague” and the invitation to the close collaboration between rulers and scientific organizations to work against fear and anxiety and the misinformation by harmful pseudoscientific newspapers represent interesting points of the manuscript.
In my opinion this Commentary has potential to be published in the present form.
Author Response
Reviewer 1
Comments and Suggestions for Authors
The Authors with the Commentary entitled "African swine fever and its natural hosts - A case study to un-ravel fake news in veterinary medicine", provide real and fact-based news on ASF, and critically elucidated the disadvantages created by internet and social media to scientists, journalists and readers, especially all fake news about human risks. In fact, the African Swine Fever (ASF), as reported by the authors, is a perfect case study to prove how fake news can undermine the public health response even in the veterinary field.
The concept of the improperly used language and the misleading word “plague” and the invitation to the close collaboration between rulers and scientific organizations to work against fear and anxiety and the misinformation by harmful pseudoscientific newspapers represent interesting points of the manuscript.
In my opinion this Commentary has potential to be published in the present form.
A: Thank you for your nice comment

Reviewer 2 Report
We are all concerned about fake news concerning human and animal diseases, and in the case of the latter it can be extremely damaging to important industries that provide livelihoods for millions of people worldwide. This paper is a great initiative and when it comes to the fake news it is timely, with some good examples of fake news relating to ASF in particular in Europe.
I attach a review that provides information on the disease in Africa to provide the kind of information that you should be familiar with in order to write about natural hosts.
There is a problem that needs to be rectified. In order to use a disease for a case study on fake news as an example, the authors need to be very well informed about the disease and have a good understanding of it that is provided by sufficient literature. The title is not entirely appropriate in stating ‘African swine fever and its natural hosts’ because there isn’t too much about the natural hosts and what there is isn’t accurate. Neither domestic pigs nor wild boars are in fact ‘natural hosts’ of the ASF virus, as they mostly get very sick and die, and the disease evolved without them. Unfortunately it appears that some if not all of the literature was just skimmed over. The result is incorrect statements and incorrect citations that will be recognisable as such to anybody with a good knowledge of the disease, including the authors of incorrectly cited literature, who are likely to be interested enough to read such an article if published. I have therefore provided an extensive review and really recommend that the authors take note and make the necessary corrections before this article is accepted for publication. If the corrections are not made, there is a real danger that the article itself will contain false if not fake information!
Line 14: Compared with many other animal diseases, ASF is contagious but not highly contagious, there is a growing body of evidence that it will usually spread slowly.
Line 15: Wild boars
Line 74: There are now 24 genotypes described, the 24th was described from Mozambique in 2018 (Quembo, C.J., Jori, F., Vosloo, W., Heath, L. 2018. Genetic characterisation of African swine fever isolates from soft ticks at the wildlife/domestic interface in Mozambique and identification of a novel genotype. Transboundary and Emerging Diseases 65, 420-431. doi: 10.1111/tbed.12700).
Line 82: Definitely not extremely contagious (Schulz, K., Conraths, F.J., Blome S, Staubach, C., Sauter-Louis, C. 2019. African swine fever: fast and furious or slow and steady? Viruses 11, 866. doi: 10.3390/v11090866; Busch, F., Haumont, C., Penrith, M.-L., Laddomada, A., Dietze, K., Globig, A., Guberti, V., Zani, L., Depner, K. 2021. Evidence-based African swine fever policies: do we address virus and host adequately? Frontiers in Veterinary Science, 8, 637487 doi: 10.3389/fvets.2021.637487).
Line 88: Definitely not boars in Sub-Saharan Africa – any wild boars (Eurasian wild boars) in Africa will be found in hunting concessions and farms as they are an exotic imported species to the continent and by the way there is no known association between wild boars and ticks (Pietschmann, J., Mur, L., Blome, S., Beer, M., Pérez-Sánchez, R., Oleaga, A., Sánchez-Vizcaíno, J.M. 2016. African swine fever virus transmission cycles in Central Europe: Evaluation of wild boar-soft tick contacts through detection of antibodies against Ornithodoros erraticus saliva antigen. BMC Veterinary Research 12:1, doi 10.1186/s12917-015-0629-9). The African wild suids (warthogs, bushpigs and an elusive creature the giant forest hog) are never called boars and should absolutely not be confused in any way at all with Eurasian wild boars. Warthogs and Ornithodoros ticks that live in their burrows are the natural and ancestral hosts of ASF virus and are impervious to any ill effects of the virus, but the whole African epidemiological situation with regard to wildlife is complex, totally divorced from the Eurasian situation because the African wild suids do not transmit the virus directly to domestic pigs and I think this is completely irrelevant to the current paper, which is about the situation of domestic pigs and wild boars, which are the same species and therefore share the same susceptibility to ASF. The reference cited is inappropriate because a vast literature is available on the ASF situation in Africa and this paper is not part of it. The only relevance of the ticks to this paper is that a species of Ornithodoros that lives in pig shelters and feeds on pigs as their preferred host became involved in the maintenance of the ASF virus and complicated control in adjacent parts of Spain and Portugal during the first important excursion of ASF outside Africa. You will find the information that you need about ticks and ASF in Europe in the following publications: Boinas, F.S., Wilson, A.J., Hutchings, G.H., Martins, C. & Dixon, L.K. 2011. The persistence of African swine fever virus in field-infected Ornithodoros erraticus during the ASF endemic period in Portugal. PLoS ONE 6(5): e20383 doi: 10.1371/journal.pone.0020383; Pérez-Sánchez, R., Astigarraga, A., Oleaga-Pérez, A., Encinas-Grandes, A. 1994. Relationship between the persistence of African swine fever and the distribution of Ornithodoros erraticus in the province of Salamanca, Spain. Veterinary Record 135(9), 207-209; Sánchez-Vizcaíno, J.M., Mur, L., Bastos, A.D.S., Penrith, M.L. 2015. New insights into the role of ticks in Africa swine fever epidemiology. Revue scientifique et technique, Office international des Épizooties 34(2), 503-511 doi: 10.20506/rst.34.2.2375.
I strongly recommend rephrasing the sentence as follows: The ASFV can spread via soft ticks (Ornithodoros genus) as was demonstrated in the Iberian Peninsula after introduction of the virus in 1960 (references as supplied).
Otherwise leave the ticks out of the equation altogether, they are by no means very widespread and most of the currently affected parts of Europe are much too cold for them to survive.
Lines 90-91: This sentence is a bit difficult to understand as it is written – I believe what it means is that where pigs are raised in outdoor systems that permit contact with wild boars, the virus can spill over to the pigs. In fact everywhere except in the current situation in Europe the disease has been driven by domestic pigs (obviously through human actions) and wild boars have become the incidental victims, but with the disease becoming established in the huge wild boar populations in much of northern Europe, it has become the other way around.
Lines 93-94: Large jumps in ASF in wild boars are definitely not ascribed to natural movement of the boars (Gilliaux, G., Garigliany, M., Licoppe, A., Paternostre, J., Lesenfants, C., Linden, A., Desmecht, D. 2019. Newly emerged African swine fever virus Belgium/Etalle/wb/2018: complete genomic sequence and comparative analysis with reference p72 genotype II strains. Transboundary and Emerging Diseases, 66(6), 2566-2591 doi: 10.1111/tbed.13302; Fekede, R.J., Wang, H., van Gils, H., Wang, X. 2021. Could wild boar be the trans-Siberian transmitter of African swine fever? Transboundary and Emerging Diseases, 68, 1465-1475 doi: 10/1111/tbed.13814. Taylor, R.A., Podgórski, T., Simons, R.R.L., Gale, S., Kelly, L.A., Snary, E.L., 2021. Predicting spread and effective control measures for African swine fever - should we blame the boars? Transboundary and Emerging Diseases, 68(2), 397-416 doi: 10.1111/tbed.13690) Gilliaux et al specifically mention than several people had been arrested in connection with the introduction into Belgium in 2018, since the nearest focus of infection was 1000 km away and boars do not usually move more than 10 km from their home range, although greater distances (but well under 100 km) have been reported (Taylor et al., 2021). There are several EFSA publications on ASF in wild boars that will bear this out. The references cited are not correct/appropriate. Reference 31 (Sauter-Louis et al) clearly states that the long-distance introductions into both Czech Republic and Belgium were due to human activity, no mention whatsoever about natural dispersal, so a clear incorrect citation that is not acceptable and the authors would very likely consider it offensive. Neither of the other two references cited even mentions long-distance jumps of the virus, they are all about carcasses. The statement about large jumps occurring due to natural movements of boars must be removed or natural movements replaced by human actions (apparently by discarding food that includes infected pig or wild boar meat where wild boars will consume it), the Sauter-Louis reference can be correctly cited for this, see Table 1. All the wild boar outbreaks are summarised and you can see if you know the background and the areas that all of the large jumps have been due to human activity (East Poland to Warsaw, Warsaw to West Poland, Hungary, Belgium…). As you will see, most of the introductions have been due to wild boar movements across borders, which were not large jumps at all, but those over longer distances were due to human activity and this is very important. The references by Fekede et al and Taylor et al confirm that wild boars do not move great distances over a short time.
Line 97: I am not sure where the ‘long asymptomatic latent period’ comes from, but the incubation period for ASF is short, 5-19 days officially but in field cases with a virulent virus like the currently circulating genotype II virus it is more likely to be 5-7 days, and a maximum of 30 days post-infection in pigs that recover. Where a long period is important is not in the live pig or boar but in uncooked pork and pork products, but one cannot really describe them as being ‘asymptomatic’.
Lines 104-107: Pigs or wild boars that recover from infection with ASF virus produce antibodies, regardless of the virulence of the virus. However, with highly virulent viruses the great majority of pigs or boars will die before antibodies become detectable, because that takes 7-11 days. That is what [31] Sauter-Louis et al. are describing. They also state that the neutralizing effect of the virus-specific antibodies is ‘controversially discussed’, so one cannot state equivocally that neutralizing antibodies are produced, the situation is a great deal more complicated than that and beyond where you need to go with this paper. Rather leave out ‘neutralizing’ in line 106 because the immune response of the pigs and boars has not been fully elucidated, which does cause problems for vaccine development.
Lines 113-124: There are many errors in this paragraph. It is very superficial and contains misinformation as a result. The errors are as follows:
Line 115: Remove ‘mainly’ – ASF evolved in eastern and southern Africa, was first described in domestic pigs in 1921 (Montgomery, 1921) and was ONLY present in eastern and southern Africa until the 1950s. The situation in Africa has changed considerably over the last decades, so to include it in the first sentence is fine.
Line 116: Again, this is not quite right. In 1957 the virus was introduced into Portugal via airport waste accessed by pigs near the airport in Lisbon and the outbreaks were rapidly eradicated, but in 1960 another ASF virus was again introduced into Portugal, apparently infecting Senegal and probably Guinea-Bissau and Cape Verde in West Africa on the way. So the authors should add ‘and again in 1960’ after 1957 to demonstrate that they do know what happened accurately.
Line 121: There is no reason whatsoever to use ‘Nevertheless’ implying a link between the European outbreaks mentioned in the preceding sentences and the current pandemic that started in 2007. The introduction of ASF into the Republic of Georgia (NOT Russia, it has been independent from Russia for decades) in 2007 was a completely separate and unrelated event, involving a different virus that came from eastern Africa (the 1957 and 1950 introductions were both from western Central Africa, almost certainly Angola). The virus was first reported in Georgia, then in Armenia and after that in dead wild boars just over the Georgian border in Chechnya, which is part of the Russian federation. This is inadequately referenced by only citing a source for outbreak history from 2016, and appropriate references are available: Rowlands, R.J., Michaud, V., Heath, L., Hutchings, G., Oura, C., Vosloo, W., Dwarka, R., Onashvili, T., Albina, E., Dixon, L.K. 2008. African swine fever isolate, Georgia, 2007. Emerging Infectious Diseases 14, 1870-1874. doi: 10.3021/eid1412.080591; Gogin, A., Gerasimov, V., Malogolovkin, A. Kolbasov, D. 2013. African swine fever in the North Caucasus region and the Russian Federation in the years 2007-2012. For a review of ASF in Europe (not absolutely accurate in every respect but nevertheless useful): Cwynar, P., Stojkov, J., Wlazlak, K. 2019. African swine fever status in Europe. Viruses, 11:310, doi: 10.3390/v11040310. A summary up to around mid-2020 is provided in Penrith, M.L. 2020. Current status of African swine fever. CABI Agriculture and Bioscience, 1, 11. doi: 10.1186/s43170-020-00011-w.
Line 122: Again not an accurate statement – while the sentence might describe the spread between 2012 and 2014 to Belarus, Ukraine, Poland and the Baltic States, its arrival in Hungary, Czech Republic and Belgium rather involved leaps and bounds (read Sauter-Louis et al. carefully, even just Table 1).
Lines 125-127: As indicated before, yes, wild boars can be responsible for transboundary spread of ASF because many of the wild boars in Europe belong to meta-populations that actually extend across borders, but it looks as if the authors just read the title and not even the abstract of the reference cited for the statement that wild boars move long distances and can therefore be responsible for transboundary spread. The authors of the paper emphasise even in the abstract that ‘long- and medium-distance spread of ASF (i.e. >30km) is unlikely to have occurred due to wild boar dispersal, due in part to the generally short distances wild boar will travel (<20 km on average). They state in the paper that boars rarely have a range of more than 10 km radius (which is quite enough to cross a border if it falls within that radius) and they cite a single report of a boar that had travelled 67 km, but that is still a very long way away from the 1000 km the boars would have to have walked to infect Belgium! The message of the Taylor et al paper in fact points out that wild boars cannot be blamed for the current situation in Europe and that human actions are more important in spreading the disease, which is true everywhere (e.g. Penrith, M.-L, Bastos, A.D., Etter, E.M.C. & Beltrán-Alcrudo, D. 2019. Epidemiology of African swine fever in Africa today: sylvatic cycle versus socio-economic imperatives. Transboundary and Emerging Diseases, 66(1), 672-686. doi: 10.1111/tbed.13117. I would actually suggest removing this paragraph as it isn’t really what the paper is about.
Line 136: High contagiousness is not the explanation for numerous outbreaks around the world, the main reason for global spread is the tenacity of the virus, i.e. its ability to survive and remain infectious for long periods in pig or wild boar meat that has not been heat treated to kill the virus. The reference emphasises the importance of this in spread and doesn’t mention ‘high contagiousness’ at all.
Lines 178-179: If the authors mean that the use of the term ‘euthanasia’ is ‘completely inappropriate’ for pigs, this is obviously wrong because especially in the EU pigs used for experimental investigation of diseases like ASF have to be euthanased when the ‘humane endpoint’ is reached, i.e. before the pig’s suffering becomes unbearable, just the same as any other animal species. The term euthanasia probably isn’t appropriate for the culling of healthy pigs to control a disease, because they are not being mercifully put out of an existing intolerable situation but are having their lives terminated as a disease control measure, often when they do not have the disease and could easily be prevented from getting it. However, euthanasia methods are used in some countries for culling operations, particularly if small numbers of pigs are involved and the aim is to avoid shedding blood, with consequent environmental contamination with the virus. An example is South Africa where pigs are sedated and then euthanased under some circumstances. Cameroon also used lethal injections for culling pigs when ASF was introduced for the first time in 1982. I agree that the statement that the Bulgarian outbreak came from Africa is fake news indeed but I think the statement that euthanasia is ‘completely inappropriate’ for pigs is also untrue and at the very least negates EU welfare regulations relating to the use of animals for research. It is even untrue for culling operations because although it is seldom used it would not be inappropriate at all, it would be ideal.
Line 184: The headline reflects a falsehood that gets repeated over and over again by the media, and the authors don’t actually comment on it – swine flu and swine fever are of course totally different diseases and the name should certainly not be used interchangeably. If I had a dollar for every time someone asked me about swine flu when they meant swine fever or had read about what was obviously swine fever but had been called swine flu, I would have quite a nice bank balance. I suggest that the authors flag this error as well.
Lines 246-248: Indeed, there was correspondence in the Lancet in 1856 sparked by a letter to the editor entitled ‘Aids and African swine fever virus’ by Feorino et al. suggesting that ASFV caused AIDS!

Author Response
Reviewer 2:
Comments and Suggestions for Authors
Q1: We are all concerned about fake news concerning human and animal diseases, and in the case of the latter it can be extremely damaging to important industries that provide livelihoods for millions of people worldwide. This paper is a great initiative and when it comes to the fake news it is timely, with some good examples of fake news relating to ASF in particular in Europe.
A1: Thank you for your appreciation and your detailed work of revision
Q2: I attach a review that provides information on the disease in Africa to provide the kind of information that you should be familiar with in order to write about natural hosts. There is a problem that needs to be rectified. In order to use a disease for a case study on fake news as an example, the authors need to be very well informed about the disease and have a good understanding of it that is provided by sufficient literature. The title is not entirely appropriate in stating ‘African swine fever and its natural hosts’ because there isn’t too much about the natural hosts and what there is isn’t accurate. Neither domestic pigs nor wild boars are in fact ‘natural hosts’ of the ASF virus, as they mostly get very sick and die, and the disease evolved without them. Unfortunately, it appears that some if not all of the literature was just skimmed over. The result is incorrect statements and incorrect citations that will be recognizable as such to anybody with a good knowledge of the disease, including the authors of incorrectly cited literature, who are likely to be interested enough to read such an article if published. I have therefore provided an extensive review and really recommend that the authors take note and make the necessary corrections before this article is accepted for publication. If the corrections are not made, there is a real danger that the article itself will contain false if not fake information!
A2: Thank you for your suggestions.
Accordingly, the title has been modified (Please, see line 2) and the entire manuscript has been revised following the comments (Please see revisions below).
Q3: Line 14: Compared with many other animal diseases, ASF is contagious but not highly contagious, there is a growing body of evidence that it will usually spread slowly.
A3: Accordingly, the manuscript has been revised (Please see line 14).
Q4: Line 15: Wild boars
A4: Accordingly, the manuscript has been revised (Please see lines 15 and 27)
Q5: Line 74: There are now 24 genotypes described, the 24th was described from Mozambique in 2018 (Quembo, C.J., Jori, F., Vosloo, W., Heath, L. 2018. Genetic characterisation of African swine fever isolates from soft ticks at the wildlife/domestic interface in Mozambique and identification of a novel genotype. Transboundary and Emerging Diseases 65, 420-431. doi: 10.1111/tbed.12700).
A5: Accordingly, the manuscript has been revised, updating the number of the genotypes (Please, see line 79) and the reference 26 has been changed as suggested (Please, see lines 379-383).
Q6: Line 82: Definitely not extremely contagious (Schulz, K., Conraths, F.J., Blome S, Staubach, C., Sauter-Louis, C. 2019. African swine fever: fast and furious or slow and steady? Viruses 11, 866. doi: 10.3390/v11090866; Busch, F., Haumont, C., Penrith, M.-L., Laddomada, A., Dietze, K., Globig, A., Guberti, V., Zani, L., Depner, K. 2021. Evidence-based African swine fever policies: do we address virus and host adequately? Frontiers in Veterinary Science, 8, 637487 doi: 10.3389/fvets.2021.637487).
A6: Accordingly, the word “extremely” has been deleted (Please, see line 82).
Q7: Line 88: Definitely not boars in Sub-Saharan Africa – any wild boars (Eurasian wild boars) in Africa will be found in hunting concessions and farms as they are an exotic imported species to the continent and by the way there is no known association between wild boars and ticks (Pietschmann, J., Mur, L., Blome, S., Beer, M., Pérez-Sánchez, R., Oleaga, A., Sánchez-Vizcaíno, J.M. 2016. African swine fever virus transmission cycles in Central Europe: Evaluation of wild boar-soft tick contacts through detection of antibodies against Ornithodoros erraticus saliva antigen. BMC Veterinary Research 12:1, doi 10.1186/s12917-015-0629-9). The African wild suids (warthogs, bushpigs and an elusive creature the giant forest hog) are never called boars and should absolutely not be confused in any way at all with Eurasian wild boars. Warthogs and Ornithodoros ticks that live in their burrows are the natural and ancestral hosts of ASF virus and are impervious to any ill effects of the virus, but the whole African epidemiological situation with regard to wildlife is complex, totally divorced from the Eurasian situation because the African wild suids do not transmit the virus directly to domestic pigs and I think this is completely irrelevant to the current paper, which is about the situation of domestic pigs and wild boars, which are the same species and therefore share the same susceptibility to ASF. The reference cited is inappropriate because a vast literature is available on the ASF situation in Africa and this paper is not part of it. The only relevance of the ticks to this paper is that a species of Ornithodoros that lives in pig shelters and feeds on pigs as their preferred host became involved in the maintenance of the ASF virus and complicated control in adjacent parts of Spain and Portugal during the first important excursion of ASF outside Africa. You will find the information that you need about ticks and ASF in Europe in the following publications: Boinas, F.S., Wilson, A.J., Hutchings, G.H., Martins, C. & Dixon, L.K. 2011. The persistence of African swine fever virus in field-infected Ornithodoros erraticus during the ASF endemic period in Portugal. PLoS ONE 6(5): e20383 doi: 10.1371/journal.pone.0020383; Pérez-Sánchez, R., Astigarraga, A., Oleaga-Pérez, A., Encinas-Grandes, A. 1994. Relationship between the persistence of African swine fever and the distribution of Ornithodoros erraticus in the province of Salamanca, Spain. Veterinary Record 135(9), 207-209; Sánchez-Vizcaíno, J.M., Mur, L., Bastos, A.D.S., Penrith, M.L. 2015. New insights into the role of ticks in Africa swine fever epidemiology. Revue scientifique et technique, Office international des Épizooties 34(2), 503-511 doi: 10.20506/rst.34.2.2375.
I strongly recommend rephrasing the sentence as follows: The ASFV can spread via soft ticks (Ornithodoros genus) as was demonstrated in the Iberian Peninsula after introduction of the virus in 1960 (references as supplied). Otherwise leave the ticks out of the equation altogether, they are by no means very widespread and most of the currently affected parts of Europe are much too cold for them to survive.
A7: Thank you for your comments and suggestions. Accordingly, the sentence about the role of the ticks has been deleted (Please, see line 88).
Q8: Lines 90-91: This sentence is a bit difficult to understand as it is written – I believe what it means is that where pigs are raised in outdoor systems that permit contact with wild boars, the virus can spill over to the pigs. In fact, everywhere except in the current situation in Europe the disease has been driven by domestic pigs (obviously through human actions) and wild boars have become the incidental victims, but with the disease becoming established in the huge wild boar populations in much of northern Europe, it has become the other way around.
A8: Accordingly, the phrase has been revised, adding the words “from domestic pigs” (Please, see line 91).
Q9: Lines 93-94: Large jumps in ASF in wild boars are definitely not ascribed to natural movement of the boars (Gilliaux, G., Garigliany, M., Licoppe, A., Paternostre, J., Lesenfants, C., Linden, A., Desmecht, D. 2019. Newly emerged African swine fever virus Belgium/Etalle/wb/2018: complete genomic sequence and comparative analysis with reference p72 genotype II strains. Transboundary and Emerging Diseases, 66(6), 2566-2591 doi: 10.1111/tbed.13302; Fekede, R.J., Wang, H., van Gils, H., Wang, X. 2021. Could wild boar be the trans-Siberian transmitter of African swine fever? Transboundary and Emerging Diseases, 68, 1465-1475 doi: 10/1111/tbed.13814. Taylor, R.A., Podgórski, T., Simons, R.R.L., Gale, S., Kelly, L.A., Snary, E.L., 2021. Predicting spread and effective control measures for African swine fever - should we blame the boars? Transboundary and Emerging Diseases, 68(2), 397-416 doi: 10.1111/tbed.13690) Gilliaux et al specifically mention than several people had been arrested in connection with the introduction into Belgium in 2018, since the nearest focus of infection was 1000 km away and boars do not usually move more than 10 km from their home range, although greater distances (but well under 100 km) have been reported (Taylor et al., 2021). There are several EFSA publications on ASF in wild boars that will bear this out. The references cited are not correct/appropriate. Reference 31 (Sauter-Louis et al) clearly states that the long-distance introductions into both Czech Republic and Belgium were due to human activity, no mention whatsoever about natural dispersal, so a clear incorrect citation that is not acceptable and the authors would very likely consider it offensive. Neither of the other two references cited even mentions long-distance jumps of the virus, they are all about carcasses. The statement about large jumps occurring due to natural movements of boars must be removed or natural movements replaced by human actions (apparently by discarding food that includes infected pig or wild boar meat where wild boars will consume it), the Sauter-Louis reference can be correctly cited for this, see Table 1. All the wild boar outbreaks are summarised and you can see if you know the background and the areas that all of the large jumps have been due to human activity (East Poland to Warsaw, Warsaw to West Poland, Hungary, Belgium…). As you will see, most of the introductions have been due to wild boar movements across borders, which were not large jumps at all, but those over longer distances were due to human activity and this is very important. The references by Fekede et al and Taylor et al confirm that wild boars do not move great distances over a short time.
A9: Thank you for your comments and suggestions. Accordingly, the phrase has been revised (Please, see lines 92-95) and suggested references have been added (Please, see references numbers 32-34 at lines 413-420).
Q10 Line 97: I am not sure where the ‘long asymptomatic latent period’ comes from, but the incubation period for ASF is short, 5-19 days officially but in field cases with a virulent virus like the currently circulating genotype II virus it is more likely to be 5-7 days, and a maximum of 30 days post-infection in pigs that recover. Where a long period is important is not in the live pig or boar but in uncooked pork and pork products, but one cannot really describe them as being ‘asymptomatic’.
A10: Thank you for your comment. Accordingly, the paragraph has been revised (Please, see lines 95-98).
Q11: Lines 104-107: Pigs or wild boars that recover from infection with ASF virus produce antibodies, regardless of the virulence of the virus. However, with highly virulent viruses the great majority of pigs or boars will die before antibodies become detectable, because that takes 7-11 days. That is what [31] Sauter-Louis et al. are describing. They also state that the neutralizing effect of the virus-specific antibodies is ‘controversially discussed’, so one cannot state equivocally that neutralizing antibodies are produced, the situation is a great deal more complicated than that and beyond where you need to go with this paper. Rather leave out ‘neutralizing’ in line 106 because the immune response of the pigs and boars has not been fully elucidated, which does cause problems for vaccine development.
A11: Accordingly, the word “neutralizing” has been deleted (Please see line 106).
Q12: Lines 113-124: There are many errors in this paragraph. It is very superficial and contains misinformation as a result. The errors are as follows:
Line 115: Remove ‘mainly’ – ASF evolved in eastern and southern Africa, was first described in domestic pigs in 1921 (Montgomery, 1921) and was ONLY present in eastern and southern Africa until the 1950s. The situation in Africa has changed considerably over the last decades, so to include it in the first sentence is fine.
A12: Accordingly, the word “mainly” has been deleted (Please see line 115).
Q13: Line 116: Again, this is not quite right. In 1957 the virus was introduced into Portugal via airport waste accessed by pigs near the airport in Lisbon and the outbreaks were rapidly eradicated, but in 1960 another ASF virus was again introduced into Portugal, apparently infecting Senegal and probably Guinea-Bissau and Cape Verde in West Africa on the way. So the authors should add ‘and again in 1960’ after 1957 to demonstrate that they do know what happened accurately.
A13: Accordingly, the phrase “and again in 1960” has been added (Please, see line 116).
Q14: Line 121: There is no reason whatsoever to use ‘Nevertheless’ implying a link between the European outbreaks mentioned in the preceding sentences and the current pandemic that started in 2007. The introduction of ASF into the Republic of Georgia (NOT Russia, it has been independent from Russia for decades) in 2007 was a completely separate and unrelated event, involving a different virus that came from eastern Africa (the 1957 and 1950 introductions were both from western Central Africa, almost certainly Angola). The virus was first reported in Georgia, then in Armenia and after that in dead wild boars just over the Georgian border in Chechnya, which is part of the Russian federation. This is inadequately referenced by only citing a source for outbreak history from 2016, and appropriate references are available: Rowlands, R.J., Michaud, V., Heath, L., Hutchings, G., Oura, C., Vosloo, W., Dwarka, R., Onashvili, T., Albina, E., Dixon, L.K. 2008. African swine fever isolate, Georgia, 2007. Emerging Infectious Diseases 14, 1870-1874. doi: 10.3021/eid1412.080591; Gogin, A., Gerasimov, V., Malogolovkin, A. Kolbasov, D. 2013. African swine fever in the North Caucasus region and the Russian Federation in the years 2007-2012. For a review of ASF in Europe (not absolutely accurate in every respect but nevertheless useful): Cwynar, P., Stojkov, J., Wlazlak, K. 2019. African swine fever status in Europe. Viruses, 11:310, doi: 10.3390/v11040310. A summary up to around mid-2020 is provided in Penrith, M.L. 2020. Current status of African swine fever. CABI Agriculture and Bioscience, 1, 11. doi: 10.1186/s43170-020-00011-w.
A14: Accordingly, the word “Nevertheless” has been deleted (please see line 121) and the paragraph has been updated using the suggested references, please see the manuscript lines 121-128 and reference numbers 41-44 (Please, see lines 443-450).
Q15: Line 122: Again, not an accurate statement – while the sentence might describe the spread between 2012 and 2014 to Belarus, Ukraine, Poland and the Baltic States, its arrival in Hungary, Czech Republic and Belgium rather involved leaps and bounds (read Sauter-Louis et al. carefully, even just Table 1).
A15: Accordingly, the phrase has been revised, describing in detail the separate events (Please, see line 121-125).
Q16: Lines 125-127: As indicated before, yes, wild boars can be responsible for transboundary spread of ASF because many of the wild boars in Europe belong to meta-populations that actually extend across borders, but it looks as if the authors just read the title and not even the abstract of the reference cited for the statement that wild boars move long distances and can therefore be responsible for transboundary spread. The authors of the paper emphasise even in the abstract that ‘long- and medium-distance spread of ASF (i.e. >30km) is unlikely to have occurred due to wild boar dispersal, due in part to the generally short distances wild boar will travel (<20 km on average). They state in the paper that boars rarely have a range of more than 10 km radius (which is quite enough to cross a border if it falls within that radius) and they cite a single report of a boar that had travelled 67 km, but that is still a very long way away from the 1000 km the boars would have to have walked to infect Belgium! The message of the Taylor et al paper in fact points out that wild boars cannot be blamed for the current situation in Europe and that human actions are more important in spreading the disease, which is true everywhere (e.g. Penrith, M.-L, Bastos, A.D., Etter, E.M.C. & Beltrán-Alcrudo, D. 2019. Epidemiology of African swine fever in Africa today: sylvatic cycle versus socio-economic imperatives. Transboundary and Emerging Diseases, 66(1), 672-686. doi: 10.1111/tbed.13117. I would actually suggest removing this paragraph as it isn’t really what the paper is about.
A16: Accordingly, the Paragraph has been deleted (Please, see line 128-136).
Q17: Line 136: High contagiousness is not the explanation for numerous outbreaks around the world, the main reason for global spread is the tenacity of the virus, i.e. its ability to survive and remain infectious for long periods in pig or wild boar meat that has not been heat treated to kill the virus. The reference emphasises the importance of this in spread and doesn’t mention ‘high contagiousness’ at all.
A17: Accordingly, the phrase has been revised, emphasizing the resistance of ASFV (Please, see lines 140-142).
Q18: Lines 178-179: If the authors mean that the use of the term ‘euthanasia’ is ‘completely inappropriate’ for pigs, this is obviously wrong because especially in the EU pigs used for experimental investigation of diseases like ASF have to be euthanased when the ‘humane endpoint’ is reached, i.e. before the pig’s suffering becomes unbearable, just the same as any other animal species. The term euthanasia probably isn’t appropriate for the culling of healthy pigs to control a disease, because they are not being mercifully put out of an existing intolerable situation but are having their lives terminated as a disease control measure, often when they do not have the disease and could easily be prevented from getting it. However, euthanasia methods are used in some countries for culling operations, particularly if small numbers of pigs are involved and the aim is to avoid shedding blood, with consequent environmental contamination with the virus. An example is South Africa where pigs are sedated and then euthanased under some circumstances. Cameroon also used lethal injections for culling pigs when ASF was introduced for the first time in 1982. I agree that the statement that the Bulgarian outbreak came from Africa is fake news indeed but I think the statement that euthanasia is ‘completely inappropriate’ for pigs is also untrue and at the very least negates EU welfare regulations relating to the use of animals for research. It is even untrue for culling operations because although it is seldom used it would not be inappropriate at all, it would be ideal.
A18: Accordingly, the word “completely” has been deleted and the phrase has been modified (Please, see line 181-184).
Q19: Line 184: The headline reflects a falsehood that gets repeated over and over again by the media, and the authors don’t actually comment on it – swine flu and swine fever are of course totally different diseases and the name should certainly not be used interchangeably. If I had a dollar for every time someone asked me about swine flu when they meant swine fever or had read about what was obviously swine fever but had been called swine flu, I would have quite a nice bank balance. I suggest that the authors flag this error as well.
A19: Accordingly, this interesting example has been added in the manuscript (Please see lines 195-198).
Q20: Lines 246-248: Indeed, there was correspondence in the Lancet in 1856 sparked by a letter to the editor entitled ‘Aids and African swine fever virus’ by Feorino et al. suggesting that ASFV caused AIDS!
A20: Accordingly, this reference (Feorino et al., 1986) has been added in the manuscript (Please see lines 258-260) and reference number 64 at lines 504-505.

Reviewer 3 Report
ASF is a highly contagious viral disease of pig that has severe impact on swine industries globally. We currently in the century of information highway. In this commentary, the authors highlighted how important it is for a news to be genuine when describing any sensitive issues like disease in social media. In this case reports on ASF was considered as an example case. Although not a lab based work, however, this is a nice case study and worth to consider for publications. My comets are as follows.
Role of wild boars in the epidemiology of ASF were discussed in two separated areas: line 91-92 and also in 125, please discuss in detail in any of these sections, not in both cases.
Number 5 heading should be “Conclusions and Recommendations”. Here recommendation could be written using bullet point. This will make the take home message easily to understand by the reader!
Need to focus more on i). the impact of these type of fake news on community ii). Importance of public awareness against such fake news using various platforms like social media-FB etc..
Author Response
Reviewer 3:
Comments and Suggestions for Authors
Q1: ASF is a highly contagious viral disease of pig that has severe impact on swine industries globally. We currently in the century of information highway. In this commentary, the authors highlighted how important it is for a news to be genuine when describing any sensitive issues like disease in social media. In this case reports on ASF was considered as an example case. Although not a lab-based work, however, this is a nice case study and worth to consider for publications. My comets are as follows.
A1: Thank you for your nice comments.
Q2: Role of wild boars in the epidemiology of ASF were discussed in two separated areas: line 91-92 and also in 125, please discuss in detail in any of these sections, not in both cases.
A2: Accordingly, the manuscript has been revised (Please see lines 92-95), also taking into account the comments of Reviewer n°2.
Q2: Number 5 heading should be “Conclusions and Recommendations”.
A2: Accordingly, the heading of the chapter 5 has been modified in “Conclusions and Recommendations” (Please see lines 305).
Q3: Here recommendation could be written using bullet point. This will make the take home message easily to understand by the reader!
Need to focus more on i). the impact of these type of fake news on community ii). Importance of public awareness against such fake news using various platforms like social media-FB etc..
A3: Thank you for your comment. Accordingly, recommendation has been divided into bullet points (Please see lines 308-315) and the paragraph has been updated.

Round 2
Reviewer 2 Report
My concerns have largely been addressed, thank you. Unfortunately reading appropriate literature, understanding its content and citing it correctly still is a problem, and therefore further corrections will be required before publication can be considered.
Lines 101-103: This paragraph still needs further correction, and was supposed to be corrected last round. As indicated in the first sentence of my comment, ALL pigs that recover from infection with any ASF virus, regardless of its virulence, produce antibodies, there are no known exceptions to that. What Sauter-Louis et al. (31) stated is that pigs infected with highly virulent strains usually die before producing detectible antibodies, which is very different from recovered pigs. They also go on to say that seropositive boars are found, meaning that they have recovered, and the majority of the strains circulating in Europe are highly virulent.
Line 116: African swine fever appeared in Sardinia for the first time in 1978 NOT 1995. Both articles cited state this very clearly, Jurado et al (39) even in the abstract so even if the authors read no further than the abstract this mistake should not have been made.
Lines 117 – 122: This revised paragraph still gets it all horribly wrong, and suggests a complete lack of knowledge of geopolitics and what countries are in Europe, also none of the countries listed between brackets as being part of the Russian Federation are part of the Russian Federation! The Baltic States and Poland are in the EU, just like Italy. Furthermore, the Gogin et al reference (42) is only about Russia and the Caucasus (Georgia, Armenia and Azerbaijan) and doesn’t mention any of the other countries for the very good reason that it was published before most of them became infected. The statement ‘then the disease was notified in Europe’ is absolute rubbish since all the preceding countries, including the affected part of the Russian Federation, are in Europe. The Cwynar reference (43) covers everything from Ukraine and Belarus to the Baltic States, Poland, and the other countries mentioned in Central and Western Europe. Finally, although the introductions into Czech Republic and northern Hungary were considered to have been due to food waste that was discarded (not necessarily illegally, a migrant worker discarding food in a garbage container is perfectly legal, just has bad results if some wild boars knock it over and eat the contents), the others either have no idea or, in the case of the Baltic States and Poland, it was due to cross-border movement of wild boars (there is relevant literature but these authors just seem to get further confused or can’t be bothered to read the literature properly. I am not sure which reference that appears under (31) is being cited for the statement about discarded food, or whether it refers to just Belgium, but please remove the statement. Gilliaux et al stated that judicial investigations had led to the detention of several people in connection with the Belgian introduction, but not what they were accused of. In line 119, 2012 and 2014 should be 2012, 2013 and 2014 – read Cwynar et al properly.
Lines 127-130: This is still a non sequitur. In the original version I had a comment that the ability of the virus to survive in meat rather than its infectiousness accounted for its spread around the world. Neither of those things have anything whatsoever to do with ASF generating fake news. That is probably due to its spectacular spread around the world and the failures of control, but the reasons WHY it could spread around the world have nothing to do with generation of fake news. It is simply a high profile disease that causes severe economic losses especially in countries that derive a lot of revenue from pork trade, so the first thing that the human species thinks is ‘Can I get it?’ and it becomes the focus of fake news. There are no technical reasons really why a disease should catch the interest of the public and become newsworthy, the main reasons would be that it makes people sick/kills them or that it has massive financial implications, making people poorer. Line 95 provides much reasons for why it generates fake news – the words ‘pandemic’ and ‘economic crisis’.
Author Response
Reviewer n° 2: Round 2
My concerns have largely been addressed, thank you. Unfortunately reading appropriate literature, understanding its content and citing it correctly still is a problem, and therefore further corrections will be required before publication can be considered.
Q1: Lines 101-103: This paragraph still needs further correction, and was supposed to be corrected last round. As indicated in the first sentence of my comment, ALL pigs that recover from infection with any ASF virus, regardless of its virulence, produce antibodies, there are no known exceptions to that. What Sauter-Louis et al. (31) stated is that pigs infected with highly virulent strains usually die before producing detectible antibodies, which is very different from recovered pigs. They also go on to say that seropositive boars are found, meaning that they have recovered, and the majority of the strains circulating in Europe are highly virulent.
A1: Accordingly, the phrase has been changed (Please, see lines 100-102).
Q2: Line 116: African swine fever appeared in Sardinia for the first time in 1978 NOT 1995. Both articles cited state this very clearly, Jurado et al (39) even in the abstract so even if the authors read no further than the abstract this mistake should not have been made.
A2: Thank you for your suggestion. Accordingly, the date has been changed from 1995 to 1978 (Please, see line 112).
Q3: This revised paragraph still gets it all horribly wrong, and suggests a complete lack of knowledge of geopolitics and what countries are in Europe, also none of the countries listed between brackets as being part of the Russian Federation are part of the Russian Federation! The Baltic States and Poland are in the EU, just like Italy. Furthermore, the Gogin et al reference (42) is only about Russia and the Caucasus (Georgia, Armenia and Azerbaijan) and doesn’t mention any of the other countries for the very good reason that it was published before most of them became infected. The statement ‘then the disease was notified in Europe’ is absolute rubbish since all the preceding countries, including the affected part of the Russian Federation, are in Europe. The Cwynar reference (43) covers everything from Ukraine and Belarus to the Baltic States, Poland, and the other countries mentioned in Central and Western Europe. Finally, although the introductions into Czech Republic and northern Hungary were considered to have been due to food waste that was discarded (not necessarily illegally, a migrant worker discarding food in a garbage container is perfectly legal, just has bad results if some wild boars knock it over and eat the contents), the others either have no idea or, in the case of the Baltic States and Poland, it was due to cross-border movement of wild boars (there is relevant literature but these authors just seem to get further confused or can’t be bothered to read the literature properly. I am not sure which reference that appears under (31) is being cited for the statement about discarded food, or whether it refers to just Belgium, but please remove the statement. Gilliaux et al stated that judicial investigations had led to the detention of several people in connection with the Belgian introduction, but not what they were accused of. In line 119, 2012 and 2014 should be 2012, 2013 and 2014 – read Cwynar et al properly.
A3: Accordingly, the paragraph has been modified (Please, see lines 113-118).
Q4: Lines 127-130: This is still a non sequitur. In the original version I had a comment that the ability of the virus to survive in meat rather than its infectiousness accounted for its spread around the world. Neither of those things have anything whatsoever to do with ASF generating fake news. That is probably due to its spectacular spread around the world and the failures of control, but the reasons WHY it could spread around the world have nothing to do with generation of fake news. It is simply a high profile disease that causes severe economic losses especially in countries that derive a lot of revenue from pork trade, so the first thing that the human species thinks is ‘Can I get it?’ and it becomes the focus of fake news. There are no technical reasons really why a disease should catch the interest of the public and become newsworthy, the main reasons would be that it makes people sick/kills them or that it has massive financial implications, making people poorer. Line 95 provides much reasons for why it generates fake news – the words ‘pandemic’ and ‘economic crisis’.
A4: This paragraph has been changed and all unnecessary info removed (Please, see lines 120-121).

Reviewer 3 Report
Thanks for all the suggested updates in this version.
Author Response
Thank you for your revision, English language has been carefully checked